# Effects of Temperature on Blood Feeding and Activity Levels in the Tiger Mosquito, *Aedes albopictus*

**DOI:** 10.3390/insects14090752

**Published:** 2023-09-08

**Authors:** Katie Costanzo, Dominic Occhino

**Affiliations:** Biology Department, Canisius University, 2001 Main St., Buffalo, NY 14208, USA; djocchin@buffalo.edu

**Keywords:** *Aedes albopictus*, behavior, blood feeding, thermoregulation, tiger mosquito, temperature

## Abstract

**Simple Summary:**

Mosquito populations experience variations in temperature across space and time as well as temperature rises associated with climate change. Our understanding of how temperature affects blood feeding behavior and activity levels in adult mosquitoes is currently limited, particularly in *Aedes albopictus*, the tiger mosquito. *Aedes albopictus* is a successful invasive mosquito around the globe that vectors several arboviruses important to human health. We evaluated the impacts of temperature on the behavior of adult females of *Aedes albopictus* from a population from St. Louis, MO. The mosquitoes were reared from hatch to adult across three temperature treatments (26 °C, 29 °C, and 32 °C) and we compared the propensity to blood feed in adults and the size of the blood meal ingested. We also measured the general activity patterns of adults that were raised for their entire life cycle across these three temperature treatments. At the highest temperature tested, female adult *Ae. albopictus* were less likely to blood feed yet moved more actively around the cage. These behavioral responses to higher temperatures suggest these conditions may be stressful to the adults, at which point they reduce normal behaviors associated with disease transmission and may seek out refugia to buffer the heat stress. These responses may negatively impact their population growth and also reduce biting and transmission rates during periods of relatively high temperatures.

**Abstract:**

Temperature has been shown to have profound effects on mosquito population dynamics and life history. Understanding these effects can provide insight into how mosquito populations and the diseases they transmit may vary across space and time and under the changes imposed by climate change. In this study, we evaluated how temperature affects the blood feeding and general activity patterns in the globally invasive mosquito species *Aedes albopictus*. We reared cohorts of *Ae. albopictus* from hatch through adulthood across three temperatures (26 °C, 29 °C, and 32 °C). The propensity of adult females to take a blood meal and the size of the blood meal were compared across temperatures. We also observed the overall activity levels of adult females over a 13.5 h period. At the highest temperature tested (32 °C), females were less likely to take a blood meal and were most active, as measured through frequency of movement. We postulate that our highest-temperature treatment imposes heat stress on adult female *Ae. albopictus*, where many abstain from blood feeding and increase movement in an attempt to escape the heat stress and find a more favorable resting location.

## 1. Introduction

As ectotherms, environmental temperature is perhaps the most important abiotic factor that affects the distribution and survival of insects. Temperature can profoundly impact the population dynamics, phenology, life history, physiology, and behavior of insects [1]. Globally, all insects are exposed to the rising temperatures associated with climate change, but many populations also experience variations in temperature across space and time within their range. An insect’s response to the environmental temperature can vary within or across species [2] and has the potential to impact the relative fitness of the individual, the dynamics of its population, and the interspecific interactions it has in the community as well [3,4,5].

Since mosquitoes serve as vectors of many human and enzootic diseases, the impacts of temperature on this insect group have been thoroughly evaluated. Temperature has been found to strongly affect mosquito survival, population growth, and several life history traits including development time, adult size, and fecundity [6,7,8,9]. Furthermore, a mosquito’s response to temperature also impacts interactions with other species including their competitors and predators [10,11], the hosts females they take a blood meal from, and the pathogens they transmit [12,13,14]. As a result, a mosquitoes’ response to temperature can also impact their vectorial capacity, or the potential to transmit disease. Traits that impact vectorial capacity include survival, population growth correlates, and host feeding behavior. Consequently, an understanding of the impacts of temperature on mosquitoes is important for our understanding of not only the ecology of these vectors but also the epidemiology of the diseases they transmit. The vectorial capacity of many mosquito populations exhibits a unimodal distribution in response to various temperatures, with a lower disease transmission at upper and lower limits within their temperature range [15].

*Aedes albopictus* (Skuse 1894), the tiger mosquito, is a globally important species both ecologically and from a standpoint of human health. Although indigenous to Asia [16], *Ae. albopictus* has been introduced to several regions globally and currently is widely distributed on nearly every continent in the world [16]. The aquatic immature stages of *Ae. albopictus* inhabit both natural and artificial containers, where it encounters and interacts with native and resident mosquito species [17]. The successful invasion of *Ae. albopictus* around the globe is attributed to the production of hardy diapausing eggs, the fast population growth rate, the often superior competitive ability in relation to other species during the larval stage, and the satyrization or mating interference that asymmetrically reduces the fitness of the resident species *Ae. albopictus* may mate with [17,18,19]. *Aedes albopictus* is also of epidemiological importance, as it can be infected with 32 arboviruses and can serve as an important vector for the transmission of human diseases such as dengue fever, chikungunya, and zika [20,21,22].

Because of the ecological and public health importance of *Ae. albopictus*, many studies have evaluated the effects of temperature on the performance of this species, particularly in the context of climate change [9]. Like other insects, *Ae. albopictus* has a unimodal response to temperature, with a wide thermal range of 10.4–40 °C to complete development, with an optimum temperature for larval development of 29.7 °C, and with lower survival at the upper and lower limits, all of which can vary across populations due to local adaptation [9,15,23]. The extensive studies evaluating the effects of temperature on *Ae. albopictus* have allowed scientists to model the populations and survival of this species across its range and predict its future distribution associated with climate change [24,25,26,27,28]. 

Although the impact of temperature on many traits in *Ae. albopictus* that influence vectorial capacity have been evaluated [9,13,29,30], to date, relatively little is known of how temperature affects the blood feeding behavior and overall adult activity patterns in this species [9]. *Aedes albopictus* is a day biter with peak feeding times in the morning and late afternoon and feeds on a wide range of hosts but often prefers humans hosts [9,23,31]. Onyango et al. [32] found that, across temperatures of 24–30 °C, female *Ae. albopictus* reduce their blood feeding frequency at the highest temperature tested. However, to our knowledge, not much else is known of the blood feeding behavior in *Ae. albopictus*, particularly with its interaction with temperature. The blood feeding activity of mosquitoes is a crucial aspect regarding the vectorial capacity through affecting the frequency contact between the mosquito vector and hosts along with infection rates [33,34]. Additionally, the size of the blood meal can affect the likelihood of taking multiple blood meals, which can increase the contact with hosts as well [33,34,35]. 

When exposed to suboptimal temperatures, behavioral responses are the primary thermoregulatory mechanism employed by ectotherms. At relatively high temperatures, many insects such as mosquitoes reduce their normal behaviors, as activity and movement can further exacerbate the heat stress [32,36,37,38]. Mosquitoes also cope with additional heat stress during the ingestion of a blood meal, which raises their body temperature at an extremely high rate. As a result, mosquitoes excrete water droplets to induce evaporative cooling and express several heat shock proteins immediately after blood feeding to reduce the damage of heat stress [39,40]. One would expect that when a mosquito is exposed to relatively high ambient temperatures, abstaining from blood feeding or ingesting smaller blood meals may inflict less additional heat stress on the mosquito.

In this study, we evaluate the effect of temperature on the blood feeding activity (propensity to blood feed and blood meal size) and overall locomotive activity over a 13.5 h observational period in adult female *Ae. albopictus*. The locomotive activity and blood feeding frequency are important in finding and feeding on a host, while the blood meal size can affect the likelihood of taking successful blood meals, all of which affect the vectorial capacity [33,41]. Each of these behaviors has the potential to alleviate or intensify the heat stress experiences of a mosquito. We tested these variables across three temperatures (26 °C, 29 °C, and 32 °C) that are commonly experienced by the population studied during the summer months when the mosquito populations are most active. We hypothesized that adult female *Ae. albopictus* would have reduced blood feeding activity (measured by the propensity to blood feed and blood meal size) and lower overall activity levels when exposed to higher temperatures as behavioral responses to reduce heat stress. 

We found partial support for our hypothesis. In our highest-temperature treatment, significantly fewer females took a blood meal compared to the other temperatures. Additionally, females moved around more frequently in the highest temperature tested. We postulate that our highest-temperature treatment imposes heat stress on adult female *Ae. albopictus*, where many abstain from blood feeding to avoid additional heat stress. We also suspect the greater frequency of movement detected in our highest-temperature treatment is repeated attempts by the mosquitos to escape the heat stress and find a more favorable resting location. Although our highest-temperature treatment is lower than the daily high temperatures this population experiences in nature, we found evidence that these ambient temperatures are indeed stressful to the adult females, leading to behavioral responses that may affect the population dynamics and transmission of the diseases they vector.

## 2. Materials and Methods

### 2.1. Propensity to Blood Feed and Blood Meal Size across Different Temperatures

We ran two blocks of an experiment to determine if temperature affects the propensity of adult *Ae. albopictus* females to blood feed and the size of the blood meal they take. Each block was performed with an F5 progeny of *Aedes albopictus* from St. Louis, MO. Eggs were hatched in deionized (DI) water in 400 mL tripour beakers with 0.25 g of nutrient broth (Difco Laboratories, Detroit, MI, USA) per liter. Following the initial setup of the hatch induction process, the tripour beakers with the eggs and nutrient broth were placed in three environmental chambers, each representing one of three of the following temperatures (26 °C, 29 °C, and 32 °C). These temperatures were selected by the average diurnal upper and lower temperature data observed from 2016 to 2018 in St. Louis, MO during the months of June–September (NOAA), when this species is established and active in the region [42].

Following hatching, 50, ~24 h larvae were transferred to 400 mL tripour beakers with 350 mL of DI water, with each beaker serving as a replicate. To provide resources, 0.0001 g per larvae of a 1:1 ratio by weight of the brewer’s yeast:lactalbumin (a total of 0.05 g) was added to each beaker. The beakers and larvae were placed back into the temperature treatment from which they hatched. Three replicates of larvae in the beakers were placed in the chambers of the three temperature treatments (26 °C, 29 °C, and 32 °C). All environmental chambers were at the L:D photoperiod of 15:9. To supplement resources throughout the experiment, an additional 0.001 g per larva of yeast:lactalbumin was added to each beaker on days 5 and 10.

Each day, each replicate across the temperature treatments was checked for pupae. Any pupae were transferred into vials with DI water until adults emerged (eclosion). Newly enclosed adults were transferred into 1 L paperboard cages with a screen mesh on top. Females that emerged from the same temperature and replicate were housed in the same cage, with up to 10 females in a single cage. To achieve 1:1 sex ratios within the cage for mating purposes, males that emerged within 5 days of the females from the same temperature treatment were placed in the same cage, with up to 20 adults in total. Cages with adults were provided constant access to cotton soaked with a 10% sugar solution for resources and housed in their respective temperature treatment. On day 8 post-emergence, the sugar solution-soaked cotton was replaced with cotton soaked with DI water only. Females were offered a blood meal on day 10 post-emergence. 

### 2.2. Blood Meal Size

Blood feeding trials were performed on day 10 post-emergence. In order to determine the size of the blood meal taken by females, we obtained a pooled weight on unfed and fed females before and after the blood feeding event, respectively. Prior to blood feeding, all mosquitoes in the cage were cold-anesthetized at approximately 7.3 °C for 3 min. Once the adults were anesthetized, the females were separated from the males from an individual cage, and their total pooled weight was estimated using an electronic balance. The average weight of unfed females from the cage was estimated. After measuring the weights, all adults were put back into their cage in their respective temperature treatment, with access to DI water for 30 min in their temperature treatment before blood feeding trials began.

For the blood feeding trials, females in a cage were provided access to a blood meal of citrated bovine blood (Hemostat Laboratories, Dixon, CA, USA) with pig intestines as an artificial membrane. Blood meals were administered with an artificial membrane feeder (Hemotek^®^, Blackburn, UK), with the temperature set at 37 °C for 30 min. Blood feeding took place inside the environmental chambers in the respective temperature treatments. We had access to two artificial membrane feeder machines and could feed up to 5 cages randomly selected from two of the three temperature treatments at the same time. Since we could not blood feed all cages on a given day concurrently, the time of the day at which the blood meals were offered was recorded for each cage. We offered blood to a total of 103 cages in the first block and 101 cages in the second block, and on any given day, we offered a blood meal from 1 to 18 cages.

Following blood feeding, the mosquitoes were cold-anaesthetized again at 7.3 °C for 3 min. Once the adults were anesthetized, the females were separated from the males within an individual cage. The number of blood-fed females was recorded. All blood-fed females were pooled together and weighed, and the average weight of the blood-fed females from a cage was estimated. Any blood-fed females were immediately sacrificed and stored at −20 °C. If, within a cage, there remained females that did not blood feed, they were returned to the cage with the males in their respective temperature treatment and provided access to a 10% sugar solution. These females were then offered an additional blood meal on day 13 post-emergence, with identical protocols as those of the first blood feeding trials.

For blood-fed females, we estimated the blood meal size relative to the female size by using the following formula:$$\mathrm{Size}\;\mathrm{of}\;\mathrm{blood}\;\mathrm{meal}/\mathrm{female}\;\mathrm{size}=\frac{\mathrm{mean}\;\mathrm{fed}\;\mathrm{female}\;\mathrm{weight}\;-\;\mathrm{mean}\;\mathrm{unfed}\;\mathrm{female}\;\mathrm{weight}}{\mathrm{mean}\;\mathrm{unfed}\;\mathrm{female}\;\mathrm{weight}}$$

To determine if the temperature affected the size of the blood meal relative to the female size, we ran a general linear mixed model (GLMM, Gaussian distribution) on the mean size of the blood meal/female size per replicate, with the temperature treatment as the fixed effect and the block and replicate as random effects (SasonDemand, SAS Institute Inc., Cary, NC, USA; PROC GLIMMIX), and we used Tukey corrections for multiple comparisons.

### 2.3. Propensity to Take a Blood Meal

After the experiment, the proportion of females that took a blood meal for each cage/treatment was calculated. We ran a separate general linear mixed model (GLMM, binomial distribution) on this variable to determine if the mean propensity to blood feed per replicate was affected by the temperature as the fixed effect and the block and replicate as random effects, with Tukey corrections for multiple comparisons. 

We ran several additional analyses to determine if our protocol introduced any confounding factors. During our protocol, in order to increase our sample size, we offered blood meals to 10-day-old females, and those females that did not blood feed were again offered a blood meal at age 13. To determine if the proportion of blood-fed females in each that blood fed the first blood feeding relative to the total females that blood fed from the first and second attempts varied across temperature treatments, we ran a general linear mixed model (GLMM, binomial distribution). We also wanted to determine if the date and time of offering a blood meal affected the proportion of females that blood fed. Because we were limited by the number of Hemotek artificial blood feeders to use, the mosquitoes were offered blood meals at different times throughout the day of the blood feeding trials. To check if our experimental protocol impacted the results, we ran a multiple linear regression (MLR) to determine if there was an effect of the date and time of day the blood meal was administered to a cage on the proportion of females that blood fed from that cage. All blood feeding trials took place between 11:30 a.m. and 4:00 p.m. (with the light period running from 6 a.m. to 9 p.m.). We systematically rotated the treatments that were fed in the earlier versus later feeding times on a given day. Although the blood source was consistent across all blood feeding trials, the age of the blood varied across the dates on which the blood meal was offered. 

For blocks I and II of the experiment, we dissected the wings of all blood-fed females and measured their length (mm) as a proxy for size across the three temperature treatments (*n* = 128). Since life history traits may impact the blood feeding behavior, we ran a Multivariate Analysis of Variance (MANOVA, PROC GLM) to determine if temperature affects the female development time and size (wing length), with Tukey corrections for multiple comparisons. For block II of the experiment, we also dissected the wings of the nonblood-fed females and ran an Analysis of Variance (ANOVA, PROC GLM) to determine if there was a difference in size (wing length) of mosquitoes that blood fed and did not blood feed across all temperature treatments in our study.

### 2.4. Mosquito Activity Levels across Different Temperatures

We ran two blocks of a separate experiment to determine if temperature affects the overall activity levels in female *Ae. albopictus*. Both blocks used an F3 progeny of *Ae. albopictus* from St. Louis, MO. Eggs were hatched using methods identical to those above in the environmental chambers across the three temperatures (26 °C, 29 °C, and 32 °C), with an L:D photoperiod of 15:9. Following hatching, 50, ~24 h larvae were added to 400 mL tripour beakers with 350 mL of DI water, with each beaker serving as a replicate, given resources identical to those identified above in the blood feeding trial experiment. The beakers with larvae were placed back into the temperature treatment from which they hatched, with four replicates per treatment.

As with the previous experiment, newly enclosed adults were transferred into 1 L paperboard cages, with females that emerged from the same temperature and replicate housed in a cage with the same number of males. Cages with adults were provided constant access to cotton soaked with a 10% sugar solution for resources and housed in their respective temperature treatment. 

On day 10 post-emergence, we randomly selected individual females and isolated them in 1 L paperboard cages with a 10% sugar solution at the bottom of the cage. The cages were returned to their temperature treatment and allowed to acclimate for at least 30 min prior to recording the behavior. Following this acclimation period, the females were recorded with a digital camcorder for 13.5 h at a standard time each day. The length of the observation period was selected based on the time needed to isolate the females and provide an acclimation period during the diurnal photoperiod time. Once the observational period began, we recorded the mosquito’s behavior until the photoperiodic lights shut off. 

The activity level was quantified by using instantaneous scan censusing, with mosquito activity observed every 10 min during the 13.5 h for which they were recorded. They were scored as “active” if they moved the distance of their body length or more from one 10 min observational scan to the next. They were scored as “changing their location” if they additionally changed their position in the cage (generally larger-scale movements) from a side of the cage to the top of the cage, from the top to bottom, from one side to another, etc., from one 10 min observational scan to the next. In the total 13.5 h for which the mosquitoes were recorded, there were 81, 10 min observational scans. The proportion of 10 min observational scans during which the mosquito was active or changed location was estimated. Our camera setup permitted the recording of one to three females in a single environmental chamber (temperature treatment) per day. In total, we recorded 50 individual female *Ae. albopictus* adults in the first block (26 °C: *n* = 15, 29 °C: *n* = 18; 32 °C: *n* = 17) and 47 adult females in the second block of the experiment (26 °C: *n* = 18, 29 °C: *n* = 15; 32 °C: *n* = 15).

We analyzed the proportion of scans for which a mosquito was scored “active” and “changed its location” using a general linear mixed model (GLMM, binomial distribution), with the treatment as a fixed effect and the block and replicate as random effects and Tukey corrections for multiple comparisons.

## 3. Results

### 3.1. Propensity to Blood Feed and Blood Meal Size across Different Temperatures

For our blood feeding trials, we had 57 females blood feed in the first block (37 females fed during the first blood feeding attempt; 20 females fed during the second blood feeding attempt) and 58 females blood feed in the second block (51 females fed during the first blood feeding attempt; 7 females fed during the second blood feeding attempt). There was no difference in the proportion of blood-fed females that fed during the first attempt across temperature treatments (GLMM F_2,14_ = 0.29, *p* = 0.75). There was also no relationship between the date (MLR: t_2,201_ = 0.26, *p* = 0.79, R^2^ = 0.003) or time of day (MLR: t_2,201_ = −0.78, *p* = 0.43, R^2^ = 0.003) at which the blood meal was offered and the proportion of females that blood fed. These results indicate that our experimental protocol of all cages not being fed concurrently and females feeding at two different ages (10 days and 13 days old) did not have any unintentional consequences regarding the results. 

Temperature affected the development time and size, as measured by the wing length of the females in our study (MANOVA F_4,250_ = 18.22, *p* < 0.0001). Females emerging from the 32 °C treatment had significantly shorter development times (12.06 ± 0.31 days) than those emerging from the 29 °C (14.31 ± 0.26 days) and 26° treatments (13.83 ± 0.21 days). There was a significant difference in the size (wing length in mm) of the adult females across all three temperatures, with the smallest females emerging from the 32°C treatment (2.46 ± 0.02 mm), larger females emerging from the 29 °C treatment (2.57 ± 0.02 mm), and the largest emerging from the 26 °C treatment (2.70 ± 0.02 mm). 

Temperature affected the propensity of females to blood feed (GLMM: F_2,10_ = 12.92, *p* = 0.0013). Significantly fewer females took a blood meal in the 32 °C treatment compared to those in the 29 °C and 26 °C treatments, while there was no significant difference between the proportion of females that took a blood meal between the 29 °C and 26 °C treatments (Figure 1). The experimental block (GLMM: F_1,10_ = 2.01, *p* = 0.16) and replicate (GLMM: F_5,10_ = 1.45, *p* = 0.29) did not affect the proportion of females that fed within a cage.

Temperature did not affect the size of the blood meal the female took relative to their body size, as measured by weight (g) (GLMM: F_2,10_ = 3.50 *p* = 0.06) (Figure 2). In block II of the experiment, the size of the females as measured by wing length, did not differ between those that took a blood meal and those that did not (ANOVA F_1,85_ = 1.98, *p* = 0.16) (mean wing length of females that blood fed: 2.6 mm ± 0.02; mean wing length of females that did not blood feed: 2.56 mm ± 0.02).

### 3.2. Mosquito Activity Levels across Different Temperatures

Temperature had a significant effect on the proportion of observational scans for which a female was active (GLMM: F_2,91_ = 7.74, *p* < 0.0008) and changed her location within the cage (GLMM: F_2,91_ = 7.54, *p* < 0.0009). Females were significantly more active and changed their location more frequently in the 32 °C treatment compared to the 26 °C and 29 °C treatments (Figure 3). Block and replicate were not significant for both the activity (GLMM: block F_1,91_ = 0.37, *p* = 0.54; replicate F_3,91_ = 0.24, *p* = 0.87) and location (GLMM: block F_1,91_ = 0.66, *p* = 0.42; replicate F_3,91_ = 1.29, *p* = 0.28).

## 4. Discussion

We found partial support for our hypotheses of reduced blood feeding activity and general locomotive activity at higher temperatures. At our highest temperature tested (32 °C), *Ae. albopictus* reduced their blood feeding activity, with significantly fewer females blood feeding compared to those in the 26 °C and 29 °C treatments. Our results are broadly consistent with a previous study [32], which found lower blood feeding rates in *Ae. albopictus* at higher temperatures (30 °C compared to 28 °C), and confirm patterns of lower blood feeding activity at even the higher temperatures (32 °C) included in our study. Overall, we had relatively low blood feeding rates, which may be affected by using artificial membrane feeders as opposed to animal hosts, but our protocols were consistent across temperature treatments. 

Blood feeding is required by most female mosquitoes to acquire the necessary rich nutrients required for egg production, but teneral lipid reserves accumulated during the larval stage can also be used to provision eggs in many species. The number of teneral lipid reserves affects the amount of blood the mosquito needs to complete vitellogenesis, egg maturation, and nonreproductive maintenance [42,43,44] and may affect the propensity to blood feed. Generally, in mosquitoes, both the temperature and resource levels during larval development can impact teneral lipid reserves [42,45], and in *Ae. albopictus*, the lipid reserves in adults are positively related to the adult body size [45,46,47]. If differences in energy reserves and physiological demands were driving the blood feeding behavior, one would expect that smaller mosquitoes with fewer teneral lipid reserves would be more likely to blood feed to enhance nutritional reserves [42]. However, in our study, the smallest adult female *Ae. albopictus* with presumably smaller energy reserves emerged in our highest-temperature treatment (32 °C), yet fewer females blood fed in the 32 °C treatment. In other studies, smaller mosquitoes were less likely to blood feed when they were derived from poorly fed larvae [48], but larvae across all of our treatments were provided with the same resource levels that were not limiting (previous studies; unpublished data). Additionally, since the mosquitoes were provided a sugar solution for 9 days post-emergence in our study, any lipid deficiencies that may have occurred within a temperature treatments during the larval stage could have been replenished. In adult mosquitoes, such access to a sugar solution for a week can substantially increase the lipid content in an adult mosquito [49,50]. Therefore, it is unlikely that the number of teneral energy reserves accumulated during the larval stage is causing the blood feeding frequency differences observed in our study, and it is more likely a result of behavioral plasticity in response to temperature. 

Many animals including insects reduce activity or normal behavior when exposed to relatively high temperatures to avoid overheating [51]. We postulate that in our highest-temperature treatment (32 °C), the adult mosquitoes are heat-stressed and respond by reducing their normal behaviors such as blood feeding. Our 32 °C exceeds the optimal temperature identified for larval development for *Ae. albopictus* (29.7 °C). For this particular species, temperature ranges of 25–30 °C are associated with the fastest development rates, greatest immature survival, highest population growth rates, shortest gonotrophic cycle, and longest adult lifespan [23,24,52,53]. Furthermore, when comparing the responses of *Ae. albopictus* when raised at 35 °C vs. 30 °C, there is a dramatic drop in the survival of immature and adult stages, along with a negative population growth rate [9,23]. Although previous studies did not specifically evaluate the responses of this species at the temperature of 32 °C, it is possible our highest-temperature treatment is putting the adults under heat stress, and reducing normal behaviors may be a mechanism for avoiding further heat stress. A reduction in activity and normal behavior patterns at higher temperatures has been observed in several arthropods, including the tick *Ixodes scapularis*, which reduces its overall movement and host seeking behaviors [54], and the fly *Bactrocera cucurbitae* delays its normal mating and oviposition behavior [55]. Our study, along with others indicate that, when exposed to high temperatures, mosquitoes reduce normal behaviors such as blood feeding and mating behaviors [32,56].

One caveat of our study is our conditions do not mimic natural conditions; in particular, we used artificial membrane feeders rather than real live hosts during our blood feeding trials. Hemotek systems with a natural membrane have been shown to be an effective artificial membrane feeding protocol in feeding rates for other species of mosquitoes [57,58] and have been used in other studies evaluating blood feeding propensity across different temperatures [32]. In our study, we fed up to 18 cages per day and chose to use artificial membrane feeders for both ethical and practical reasons. However, this protocol does not mimic the natural conditions of a real live host. Mosquitoes use several cues to detect hosts, including carbon dioxide, temperature, and odors from the host and its microbiome itself [59]. The hemotek system predominately attracts mosquitoes through the temperature stimulus, and the use of natural pig intestine membrane may produce chemical attractants as well. However, the mosquito’s ability to detect the temperature stimulus of the hemotek may vary across our temperature treatments. Although mosquitoes have been found to be extremely sensitive to temperature and are able to detect a temperature host stimulus of 2.5 °C above the ambient temperature, with temperatures of 5 °C above ambient temperatures eliciting host-seeking behaviors, the various temperature differentials between the artificial membrane feeder and ambient temperature across treatments may have impacted the results [60]. As such, further studies that validate the results of our study and others [32] should use live hosts such as anesthetized mice. 

In our locomotive activity study, over the entire 13.5 h observational period, adult female *Ae. albopictus* spent relatively little time engaged in activity (<15–25% of observations). The relatively low activity levels across all temperature treatments confirm that mosquitoes including *Ae. albopictus* spend the majority of their time resting [61,62]. 

When exposed to high temperatures that induce stress, the most common and effective behavioral thermoregulatory mechanisms in ectotherms such as insects are a reduction in the amount of activity, shifting the timing of activity, and dispersal to cooler microhabitats to reduce heat exposure [37,63,64]. In our study, we hypothesized that *Ae. albopictus* would reduce their overall activity when exposed to relatively high temperatures to reduce further heat stress; however, we found the opposite to be the case, with *Ae. albopictus* being most active and moving most frequently in our highest-temperature treatment. We postulate that the elevated activity levels observed in our highest-temperature treatment is indicative of the mosquitoes attempting to escape the unfavorable temperature conditions in this treatment. 

Dispersing to a more favorable microhabitat can be an effective tactic for reducing heat stress, as temperatures across microhabitats can vary tremendously and act as a buffer [65,66,67,68]. For instance, many insects will move towards shaded or cooler habitats towards the soil, which allow them to survive in an otherwise lethal environment [60,65,67,69]. Mosquitoes will commonly rest in well-shaded crevices of earth banks, stones and trees, or within vegetation or human structures as a refuge [61] (pp. 316–317). Adult mosquitoes of the genus *Aedes* most frequently rest in vegetation closer to the ground and in the shade and avoid resting sites with temperatures above 30 °C [69,70,71], all of which are behavioral thermoregulatory mechanisms. 

However, heterogeneity must exist in the environment for shade seeking and microhabitat use to buffer the heat stress associated with high temperatures. The cages used in our study did not include any microhabitat heterogeneity, shade, or refugia from the temperature. The higher activity observed in these treatments could be the repeated attempts of the adult female *Ae. albopictus* to escape the stress under the continued high temperatures they were exposed to. When subjected to high temperatures, other insects will increase activity to escape the stressful environment until that refugia is encountered [65,72,73]. The lack of microhabitats provided in our study may have induced more frequent movement in adult *Ae. albopictus* females as an attempt to escape the heat stress they were exposed to. When mosquitoes are provided with microhabitats that differ in temperature, they generally prefer areas with cooler temperatures, particularly when the ambient temperature is relatively high, and will avoid temperatures above 30 °C [38,74,75]. It would be of interest to perform additional studies evaluating the activity levels in adult *Ae. albopictus* and refugia use across different temperature regimes to remove any confounding effects of possible repeated attempts to seek out refuge that was available.

The temperatures tested in our study (26 °C, 29 °C, and 32 °C) were selected from the average upper, mid, and lower diurnal temperatures recorded from 2016 to 2018 in St. Louis, MO during the months of June–September (NOAA) to evaluate how the population of *Ae. albopictus* responds to temperatures it commonly encounters in nature. Although we found evidence that at the highest temperature tested, *Ae. albopictus* shows signs of thermal stress behaviorally, on a given day, the populations may experience temperatures exceeding those of our high-temperature treatment. During the months of June–September from 2016 to 2018 in St. Louis, from which we selected our temperature treatments, there were 9–23 days within a month on which the high temperature exceeded that of our temperature of 32 °C. These high temperatures ranged from moderate (34.4 °C) to extreme (42.2 °C) temperatures above our high-temperature treatment. Furthermore, average high temperatures will continue to rise as a result of climate change (IUCN). Therefore, these populations must cope with greater temperature stress than what we evaluated and temperatures above what has been identified as the thermal max for the larval development of this species (40 °C) [23]. Since they are periodically exposed to temperatures at and above our highest-temperature treatment in nature, they likely seek refuge and reduce normal behaviors such as blood feeding during these times. The use of refugia along with reduced blood feeding activity could affect their population growth through reduced mating opportunities and egg production. Additionally, there would be a reduction in the contact with hosts and a potentially reduced vectorial capacity under periods with relatively high temperatures. At a relatively high temperature of 30 °C (below our highest-temperature treatment), *Ae. albopictus* had a reduced total vectorial capacity for Zika [32]. 

We implemented a constant temperature in our experimental treatments to specifically evaluate how mosquitoes respond to particular temperatures. However, in nature, the daily temperature fluctuates throughout the day, with more and less favorable periods with respect to temperature. *Aedes albopictus* has peak feeding periods in the morning and late afternoon, which may correspond to favorable environmental conditions and host activity [23,31]. On stressfully hot days, females may not only shift their position spatially to a microhabitat that may reduce heat stress, but they also may shift their host feeding times to more favorable periods of the day. The relative length of time with favorable temperatures will vary from day to day and may be dramatically reduced during periods of heat waves.

Behavioral mechanisms are the most common and most effective thermoregulation mechanisms in insects [36,37,63,64], yet little is known of how mosquitoes behaviorally respond to the heat stress associated with high ambient temperatures. Our study suggests that adult female *Ae. albopictus* reduce normal behaviors such as blood feeding but increase the frequency of movement, possibly attempting to seek out refugia or more moderate environments. Further studies are needed to evaluate the impact of temperature on the microhabitat seeking and activity levels of this species while in a heterogenous environment and evaluate these responses under even higher temperatures that they experience in nature.

## Figures and Tables

**Figure 1 insects-14-00752-f001:**
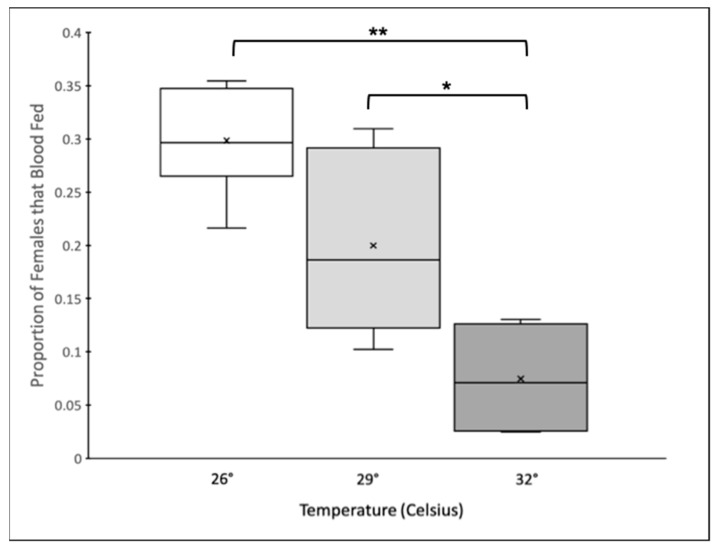
Box and Whisker plot of the proportion of females that blood fed within an individual cage across temperature treatments. The center line indicates the median (50th percentile), the boxes contain the 25th to 75th percentiles, and the ×s represent the mean. Boxplots connected by brackets and asterisks indicate pairwise comparisons between means that are significantly different (* = *p* < 0.01, ** = *p* < 0.001).

**Figure 2 insects-14-00752-f002:**
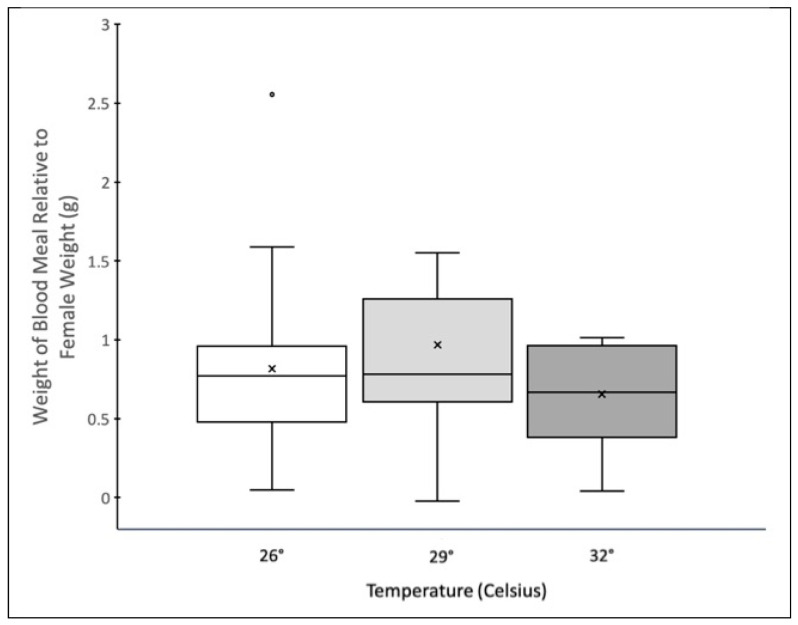
Box and Whisker plot of the weight of the blood meal ingested relative to the weight of the female (see methods for the equation). The center line indicates the median (50th percentile), the boxes contain the 25th to 75th percentiles, dots represent outliers and the ×s represent the mean.

**Figure 3 insects-14-00752-f003:**
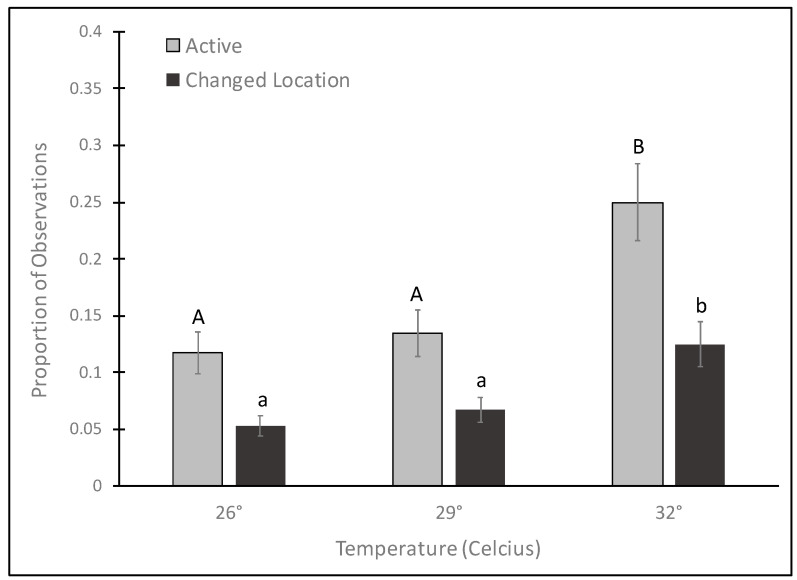
Mean proportion (±1 standard error) of 10 min observations over a 13.5 h period during which female *Ae. albopictus* adults were active (when they moved the length of their body length or more) or changed location (when they changed location in the cage from one side to another, from a side to the bottom, etc.). Means with different capital letters indicate significant pairwise differences between the proportion of observations that mosquitoes were active. Means with different lowercase letters indicate significant pairwise differences between the proportion of observations that mosquitoes changed location within the cage.

## Data Availability

Data are available upon request.

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
