# Peer review of "Effects of Temperature on Blood Feeding and Activity Levels in the Tiger Mosquito, Aedes albopictus"

_insects, 2023, doi:10.3390/insects14090752_

Round 1

Reviewer 1 Report

In this study, the authors investigate the effects of three temperature regimes (26, 29 and 32C) on the proportion of Aedes albopictus mosquitoes feeding on blood via a membrane feeding device and the amount of blood they ingest. They also measure the activity of individual mosquitoes at these temperatures. Their experiments are straightforward but there are some key methodological flaws which introduce confounding factors and make the results difficult to draw any solid conclusions from.

The authors use membrane feeders which are not recommended for behavioral studies due to their innate variability and the lack of important host cues presented in a live host such as body odors and CO2. Having a higher ambient temperature could make it more difficult for mosquitoes to sense the heat of the warm membrane feeder. As the authors only measure blood feeding through counting engorged mosquitoes, they are unable to explain if their results are due to a lack of attraction at higher temperatures or a lower ability to feed on the membrane surface, and so on. The feeding proportions are also quite low (below 30% in all treatments) which means that are clearly other factors beyond temperature influencing feeding.

The authors calculate blood meal weight relative to mosquito weight by weighing groups of mosquitoes prior to feeding, feeding the mosquitoes blood, then weighing the mosquitoes that fed. Weighing mosquitoes in groups is problematic because the number of mosquitoes in the pre-fed and post-fed groups will differ if not all mosquitoes feed. In fact, due to the low feeding proportions in this study (below 30% in all treatments), most mosquitoes weighed to calculate unfed weight are not being weighed again to measure fed weight, yet the latter measurement is divided by the former measurement to calculate relative blood meal weight. If there are any differences in size between the mosquitoes that fed and those that did not feed, this will bias the results.

The authors mention in the methods section that they took mosquitoes that did not feed in the initial experiment, returned them to cages, then fed them again later. It is unclear how many of these cages were included in the dataset but this is clearly a substantial confounding factor. Not only were the mosquitoes a different age, but the fact they didn’t feed the first time means that this pool of mosquitoes represents a group where the most “fit” mosquitoes have been removed, so the remaining group is likely to differ in a number of fundamental ways such as body size.

Mosquitoes in each temperature treatment are both reared at the given temperature and also blood fed at that temperature. This design means that the authors are unable to separate these two effects, leading to their conclusions containing much speculation about whether the reduced blood feeding and changes in activity are due to the smaller size of the mosquitoes or if it’s a behavioural response to temperature (or both). This could be avoided by including a temperature during development or blood feeding, and having a common temperature at other times, or even performing a reciprocal treatment where mosquitoes reared at all temperatures are then blood fed at a range of temperatures.

Line 12-13 – “Temperature treatment” is a bit vague- it is worth emphasizing that the mosquitoes in each treatment were both reared at and fed at the given temperature

Line 106 – Provide units for 13.5 – is this 13.5 hours?

Line 192 – There is a formatting issue with the equation here

Line 253 – Is this 50 individuals in total, or 50 per temperature treatment?

Line 262 – Indicate the statistical test used when showing stats as different tests are used throughout the paper

Figures – Use boxplots or scatter dot plots  – bar charts provide no information about the distribution of the data and are no longer commonly used for this type of data

Line 274 – Significantly different by which statistical test?

Line 275-277 – These two statements contradict each other. There is also an apparent contradiction between the general linear mixed model stats and the figure – there is no significant effect of temperature in the text but there are significant differences in the figure

Line 321 –  The authors do not test blood feeding frequency here, only the proportion of mosquitoes that fed within 30 minutes, so the results here don’t appear to be inconsistent with these predictions. Testing this prediction would require a different set of experiments with multiple opportunities for blood feeding.

Lines 347-350 – There is too much of an emphasis on significance here and a reiteration of the results. It also repeats conflicting statements that there is no significant effect on blood meal size yet there are significant differences between temperatures. The discussion should instead focus on the implications of these findings.

Line 360 – The figure does not indicate variability – having a dot plot or boxplot would make this variability much clearer

Reviewer 2 Report

This manuscript describes a series of experiments investigating the influence of temperature on Aedes albopictus blood feeding, activity level, and body size in a highly controlled laboratory environment. There is a sizable body of literature on this topic, however, many of the published studies investigate these parameters at extreme temperatures to identify thermal limits to various activities or survival.  Here, the authors investigate the proportion of mosquitoes taking a blood meal and the size of the blood meal relative to the size of the female mosquito over a relatively small temperature range (26C – 32C) that is more representative of the temperatures experienced in the field by the Ae. albopictus strain used in this study.  The experimental design is thoroughly described in the Methods section and is appropriate to address the questions posed in the study.  The results are clearly presented and the statistical analyses appear to be appropriately applied. Both the Introduction and Discussion are thoroughly documented with appropriate citations.  The conclusions and relevance to the broader field are clearly stated and supported.

 I have only three minor items for the authors to consider:

Age of the mosquitoes when first offered the blood meal:  As described in the Methods, female mosquitoes were held with free access to 10% sucrose solution for 9 days and water only for 1 day prior to being evaluated for blood feeding activity.  While this is common practice to maximize blood feeding in mosquito laboratory colonies, it likely does not reflect what occurs in the field where female mosquitoes likely start blood-feeding activity within the first several days after emergence, and without constant access to an exogenous energy source.  It is possible that access to sugar for such a long time maximizes energy reserves that are not depleted in the 24 hours of sugar deprivation (see Van Handel, E, (1965), The obese mosquito. The Journal of Physiology, 181 doi: 10.1113/jphysiol.1965.sp007776.) and may influence blood feeding behavior (see: Nasci RS. Influence of larval and adult nutrition on biting persistence in Aedes aegypti (Diptera: Culicidae). J Med Entomol. 1991 Jul;28(4):522-6. doi: 10.1093/jmedent/28.4.522. PMID: 1941913.)  While this does not negate the observations in this manuscript, it is probably worth addressing in the discussion.

 Lines 192-194:  The formula for estimating blood meal size relative to female size is not formatted properly in the PDF version I received for review, and needs to be edited.

 Lines 70-72:  While all of the viruses listed have been isolated from Aedes albopictus, it is only considered an important vector of dengue, chikungunya and zika viruses.  There is no good evidence that it contributes in an important way to the public health risk from La Crosse, eastern equine encephalitis or West Nile virus.

Reviewer 3 Report

 The paper is interesting, written well and statistical analyses were sufficient. Nevertheless, there are some issues with this study and some others that attempt to assess the impact of temperature (or even more, by emulating the process of global warming) on the life history traits of mosquitoes. The primary concern that is frequently overlooked pertains to the fluctuation of daily temperature. At higher temperature larvae will be happily raising, however, the adults at lab conditions will be really stressed, seeking for cooler places. At higher temperatures, larvae will be happily raising, however, adults at lab conditions will be stressed looking for cooler places.

 In this paper, the authors raised the larva at three different temperatures, which were always constant. The authors mentioned that these three temperatures were the average temperatures during the day in St. Louis from 2016 to 2018. However, this does not mean that the temperatures were constant throughout the day. What is more, these temperatures were not necessarily registered during the “preferring feeding” hours of Ae albopcitus. Regardless of the water and ambient temperature being the same, the thermal water conductivity is completely different. Despite the well-studied effect on larval development due to water temperature, the artificial exposition at the highest temperatures will not necessarily reflect the actual feeding and movement mosquito behavior. It is important to note that mosquitoes usually have preferred feeding times, but this does not mean that they cannot try to feed at different times, especially if weather limits their behavior. Just imagine when it is raining at the “preferred feeding” hours. We used the term 'preferred feeding hours,' but we actually mean the hours when weather conditions are most suitable for performing a behavior.

With the above paragraphs context, I should ask the authors:

The biological reasons behind the idea of raising larvae at three but different (and constant) temperatures and then exposed the adults the same temperature.

Why not expose the adults to the other two temperatures.

Did water temperatures had an effect on development time?

So, it is possible that your study was not that artificial and did not reflect the real behavior at the field?

Please do not misunderstand, I am confident that you are in the right mind. I am merely attempting to convey the limitations of your research (and other studies) through biological reasoning.

Round 2

Reviewer 1 Report

I thank the authors for including additional data and analyses to address the potential confounding factors I raised in the previous review.

While I still have reservations about the experimental design, I think the authors have sufficiently addressed or acknowledged these key limitations of the study.

Author Response

The reviewer was satisfied with our revisions and did not have any additional queries to address.

Reviewer 3 Report

The authors have done a nice job of revising this manuscript and addressing the reviewer comments and queries

Author Response

The reviewer was satisfied with our revisions and did not have any more additional queries to address.